

# The impact of skeletal muscle index at the third lumbar spine on nosocomial deterioration and short-term prognosis in acute pancreatitis: a retrospective observational study

Chuntao Lin[1,*], Junyuan Zhang[2,*], Chunye Wang[1], Wei Lian[1] and Yicong Liu[2]

[1] Department of Radiology, Yantaishan Hospital, Yantai, China
[2] Department of Radiology, Yantai Yuhuangding Hospital, Yantai, China
* These authors contributed equally to this work.

Corresponding author
Yicong Liu, 15953598828@163.com

## ABSTRACT

**Objective:** To investigate the impact of the third lumbar skeletal muscle index (L3-SMI) assessed by CT on the in-hospital severity and short-term prognosis of acute pancreatitis.

**Methods:** A total of 224 patients with severe acute pancreatitis admitted to Yantaishan Hospital from January 2021 to June 2022 were selected as the subjects. Based on the in-hospital treatment outcomes, they were divided into a mortality group of 59 cases as well as a survival group of 165 cases. Upon admission, general information such as the Acute Physiology and Chronic Health Evaluation II (APACHE II) score, along with the abdominal CT images of each patient, were analyzed. The L3-SMI was calculated, and the Modified CT Severity Index (MCTSI) and Balthazar CT grade were used to assess the severity of in-hospital complications of acute pancreatitis. The evaluation value of L3-SMI for the prognosis of severe acute pancreatitis was analyzed, as well as the factors influencing the prognosis of severe acute pancreatitis.

**Results:** No statistically significant differences in gender, age, BMI, etiology, duration of anti-inflammatory drug use, and proportion of surgical patients between the survival and mortality groups were observed. But the mortality group showed higher proportions of patients with an elevated APACHE II score upon admission, mechanical ventilation, and renal replacement therapy, compared to the survival group, with statistically significant differences ($P < 0.001$). Furthermore, the mortality group had higher MCTSI scores ($6.42 \pm 0.69$) and Balthazar CT grades ($3.78 \pm 0.45$) than the survival group, with statistically significant differences ($P < 0.001$). The mortality group also had a lower L3-SMI ($39.68 \pm 3.25$) compared to the survival group ($42.71 \pm 4.28$), with statistically significant differences ($P < 0.001$). L3-SMI exhibited a negative correlation with MCTSI scores and Balthazar CT grades ($r = -0.889, -0.790, P < 0.001$). Logistic regression analysis, with mortality of acute pancreatitis patients as the dependent variable and MCTSI scores, Balthazar CT grades, L3-SMI, APACHE II score upon admission, mechanical ventilation, and renal replacement therapy as independent variables, revealed that MCTSI scores and L3-SMI were risk factors for mortality in acute pancreatitis patients ($P < 0.001$).

Logistic regression analysis using the same variables confirmed that all these factors were risk factors for mortality in acute pancreatitis patients.

**Conclusion:** This study confirmed that diagnosing muscle depletion using L3-SMI is a valuable radiological parameter for predicting in-hospital severity and short-term prognosis in patients with acute pancreatitis.

# INTRODUCTION

Acute pancreatitis is one of the most common clinical diseases, with a global annual incidence being 34 (95% confidence interval: 23–49) cases per 100,000 people (*Petrov & Yadav, 2019*). While most cases of acute pancreatitis are self-limiting with a good prognosis, severe acute pancreatitis can lead to the release of a large number of inflammatory mediators and cytokines following organ damage caused by pancreatic enzyme entry into the bloodstream. This severity stems from pancreatic enzymes entering the bloodstream, causing organ damage and releasing vast amounts of inflammatory mediators and cytokines. Consequently, early systemic complications such as systemic inflammatory response syndrome and multiple organ dysfunction syndrome can arise as a result. Mortality rates from severe acute pancreatitis can range from 36% to 50% (*Banks et al., 2013*). Therefore, a precise assessment of disease severity and prognosis on admittance is crucial for effective patient management.

While risk stratification and individualized treatment procedures can enhance patient outcomes, established evaluation methods, such as CT with a quantitative analysis of pancreatic and adjacent tissue necrosis, have limitations (*He et al., 2022*). Evidence by *Zhou et al. (2023)* suggests that between 60% and 85% of acute pancreatitis patients suffer from malnutrition to some degree. This significantly impacts the treatment's effectiveness and the patient's prognosis (*Zhou et al., 2023*), underscoring the importance of a reliable and accurate assessment of the patient's nutritional status.

Recently, several hematological indicators and scoring tools like the Prognostic Nutritional Index (PNI) and the Controlling Nutritional Status (CONUT) have emerged as dependable measures of the body's nutritional reserves (*Lee et al., 2021*). Simultaneously, scholars have begun leveraging computerized tomography (CT) technology to provide a L3-Skeletal Muscle Index (L3-SMI) at the third lumbar vertebra. This measure aids in quantifying systemic muscle quality and quantity, thus assessing the patient's nutritional status.

Evidence confirms that the L3-SMI is a promising approach to predict hospitalization prognosis. Moreover, it's noted that a low L3-SMI constitutes an independent predictive parameter for adverse outcomes in acute pancreatitis patients. However, factors such as ethnicity, age, gender, obesity, and treatment methods influence the benchmarks for muscle depletion and malnutrition assessment using L3-SMI (*Lizan, Perez-Carbonell & Comellas, 2021*).

Regrettably, in our locale, there remains a paucity of research underscoring the prognostic value of L3-SMI in acute pancreatitis patients. This study, therefore, seeks to illuminate the relationship between the L3-SMI, as ascertained from abdominal CT scans, and the prognosis of patients with acute pancreatitis.

## METHODS

### General data

This study was a retrospective observational study. Based on the in-hospital treatment outcomes, the patients were divided into two groups: a mortality group, consisting of 59 cases where the patients died within 7 to 30 days after admission, and a survival group (all patient receive recovery and leave hospital) (*Fonseca Sepúlveda & Guerrero-Lozano, 2019*), consisting of 165 cases. All samples obtained in this study were approved by the ethics committee of the Yantaishan Hospital and abided by the ethical guidelines of the Declaration of Helsinki, and ethics committee agreed to waive informed consent.

The inclusion criteria were as follows: 1. meeting the diagnostic criteria for severe acute pancreatitis as outlined in the "China Acute Pancreatitis Diagnosis and Treatment Guidelines (2019, Shenyang)" (*Lorenz et al., 2023*); 2. complete clinical and imaging data records; 3. patient age >18 years, with symptom onset within 3 days; 4. no anticoagulant medication taken within the past month.

The exclusion criteria were as follows: 1. a history of stroke or acute myocardial infarction; 2. presence of hematological disorders; 3. severe organ dysfunction such as heart, liver, kidney, lung, or concomitant tumors; 4. pregnancy or lactation in women; 5. patients treated by doctors not in the authors list.

General information like gender, age, etiology, body mass index (BMI), duration of anti-inflammatory drug use, surgical history, Acute Physiology and Chronic Health Evaluation II (APACHE II) score upon admission, mechanical ventilation status, and renal replacement therapy of all critically ill patients with acute pancreatitis were collected and organized.

### Abdominal CT examination

Within 48 h of admission, abdominal CT examinations were performed using a CT scanner with the following parameters: tube current of 100 mA, tube voltage of 120 kV, pitch of 1.484:1, slice thickness of 5 mm, covering a region from 2 cm above the diaphragm muscle to the anterior superior iliac spine. The procedure began with a routine plain scan, followed by the administration of a non-ionic iodinated contrast agent (Ultravist) for contrast-enhanced scanning. The injection rate was set at 2.5–3.5 mL/s, with a total volume of 80–100 mL. The arterial phase was scanned at 30–35 s, the portal venous phase at 60–70 s, and the delayed phase at 160–180 s. Each scan was completed during a single breath-hold. All of the aforementioned examination procedures were performed by the same physician. Two radiologists from the imaging department independently reviewed the images and conducted a comprehensive assessment.

## Measurement of L3-SMI

The abdominal CT images obtained at the time of admission for each patient were retrieved. Based on the different Hounsfield values, a region of interest was segmented at the L3 level to determine the extent of the surrounding muscles, including the psoas major, erector spinae, rectus abdominis, quadratus lumborum, transverse abdominis, external and internal obliques. The average cross-sectional area of these muscles (cm$^2$) was then calculated. Subsequently, the average cross-sectional area of the L3-level muscles (cm$^2$) was standardized by dividing it by the square of the patient's height (m$^2$) to calculate the L3-SMI (cm$^2$/m$^2$).

## Modified CT severity index and Balthazar CT grade assessment

The modified CT severity index (MCTSI) score (ranging from 0 to 10) and the Balthazar CT grade (ranging from 0 to 4) were used for evaluation. The MCTSI score was determined based on the degree of pancreatic inflammation (0 points for normal, two points for pancreatitis and peripancreatic inflammation, and four points for the presence of fluid collection or peripancreatic fat necrosis), the extent of necrosis (0 points for no necrosis, two points for ≤30% necrosis, and four points for >30% necrosis), and the presence of extrapancreatic complications (two points for complications such as ascites, pleural effusion, gastrointestinal or vascular involvement) (*Meure, Steer & Porter, 2023*). The Balthazar CT grade assessment was categorized as follows: Grade A (0 points) for a normal pancreas, Grade B (one point) for pancreatic parenchymal changes including localized or diffuse gland enlargement, Grade C (two points) for peripancreatic and pancreatic parenchymal inflammatory changes with mild peripancreatic exudation, Grade D (three points) for significant peripancreatic exudation or the presence of a single fluid collection within the pancreatic parenchyma or peripancreatic area, and Grade E (four points) for extensive intra- and extra-pancreatic fluid accumulation, including pancreatic abscess, fat, and pancreatic necrosis (*Nyirjesy et al., 2023*).

## Statistical analysis

The statistical analysis was carried out with the aid of SPSS 25.0 software. Numerical data were tested for normality, which was confirmed using a Shapiro-Wilk test. Continuous variables were reported as mean ± standard deviation. Then, the group differences were further assessed using the two-sample independent test. Categorical variables were showed as frequencies as well as its percentages. Then, its group differences were analyzed using the chi-square test. The receiver operating characteristic (ROC) curve analysis was conducted to evaluate the predictive ability of the L3-Skeletal Muscle Index (L3-SMI) for the prognosis of severe acute pancreatitis. The area under the curve (AUC) for adverse prognosis corresponding to L3-SMI was calculated to assess the diagnostic value. Additionally, the optimal threshold value for L3-SMI, along with its associated sensitivity and specificity, was determined. Spearman's correlation test was applied to analyze the correlation between variables. Multivariable logistic regression analysis was run to identify factors influencing the prognosis of acute pancreatitis. MCTSI score, Balthazar CT grade,

**Table 1 A comparison of clinical data between the survival group and the group that experienced mortality.**

| Variables | | Survival group ($n = 165$) | Non-survivors group ($n = 59$) | $t/\chi^2$ | $P$ |
|---|---|---|---|---|---|
| Age (years) | | 56.28 ± 5.63 | 57.79 ± 5.24 | 1.800 | 0.073 |
| Male (cases, %) | | 96 (58.18) | 42 (71.19) | 3.107 | 0.078 |
| BMI (kg/m²) | | 24.17 ± 2.45 | 24.42 ± 2.57 | 0.664 | 0.507 |
| Etiology (cases, %) | Cholangiogenic | 55 (33.33) | 19 (32.20) | 0.115 | 0.944 |
| | Alcoholic | 63 (38.18) | 24 (40.68) | | |
| | Hyperlipidemic | 47 (28.49) | 16 (27.12) | | |
| Duration of anti-inflammatory drug usage (days) | | 23.17 ± 5.28 | 29.03 ± 5.54 | 7.222 | <0.001 |
| Admission APACHE II score | | 13.06 ± 4.74 | 14.12 ± 5.03 | 1.451 | 0.148 |
| Surgery (cases, %) | | 49 (29.70) | 17 (28.81) | 0.016 | 0.898 |
| Mechanical ventilation (cases, %) | | 138 (83.64) | 58 (98.31) | 8.550 | 0.004 |
| Renal replacement therapy (cases, %) | | 23 (13.94) | 22 (37.29) | 14.759 | <0.001 |

Note:
BMI refers to Body Mass Index, and APACHE II stands for Acute Physiology and Chronic Health Evaluation II Score. Numerical data were tested for normality, which was confirmed using a Shapiro-Wilk test. Continuous variables were reported as mean ± standard deviation. Then, the group differences were further assessed using the two-sample independent test. Categorical variables were showed as frequencies as well as its percentages. Then, its group differences were analyzed using the chi-square test.

L3-SMI, admission apacheii, mechanical ventilation, and renal replacement therapy were used to established multivariable logistic regression analysis. The doorway for statistical significance was established at two-sided $P$-values < 0.05.

# RESULTS

## The clinical data for survival as well as mortality group

A total of 270 patients initially met the eligibility criteria for inclusion in the study. However, 46 patients were excluded based on the predefined exclusion criteria. Finally, a total of 224 critically ill patients with acute pancreatitis who were admitted to Yantaishan Hospital from January 2021 to June 2022 were selected as the subjects of this study. The patients' ages ranged from 45 to 82 years, with 138 males and 86 females.

In the selection of the study cohort, it is pertinent to note that a subset of patients underwent surgical interventions as part of their acute pancreatitis management. Among the 224 critically ill patients with acute pancreatitis included in the study, 66 individuals had undergone surgical procedures as a component of their treatment.

There were no statistically significant differences between the survivor as well as non-survivor groups in terms of sex, age, BMI, etiology, duration of anti-inflammatory drug use, and proportion of surgical patients. However, the non-survivor group had significantly higher admission scores for APACHE II, proportion of mechanically ventilated patients, and proportion of patients receiving renal replacement therapy compared to the survivor group (Table 1).

## Comparison of MCTSI score, Balthazar CT grade, and L3-SMI between survivors as well as non-survivors

The comparison between the survival and non-survivor groups revealed statistically significant differences in the mean scores for MCTSI and Balthazar CT Grade, with higher

**Table 2 Comparison of MCTSI score, Balthazar CT grade, and L3-SMI between the survival group and the death group (mean ± SD).**

| Group | n | MCTSI score | Balthazar CT grade | L3-SMI (cm²/m²) |
|---|---|---|---|---|
| Survival group | 165 | 5.27 ± 0.58 | 3.27 ± 0.38 | 42.71 ± 4.28 |
| Non-survivors group | 59 | 6.42 ± 0.69 | 3.78 ± 0.45 | 39.68 ± 3.25 |
| t | - | 13.012 | 8.416 | 4.949 |
| P | - | <0.001 | <0.001 | <0.001 |

scores observed in the non-survivor group (MCTSI: 6.42 ± 0.69 *vs.* 5.27 ± 0.58; Balthazar CT Grade: 3.78 ± 0.45 *vs.* 3.27 ± 0.38, both $P < 0.001$). Additionally, the third lumbar skeletal muscle index (L3-SMI) displayed a lower mean value in the non-survivor group compared to the survival group (39.68 ± 3.25 *vs.* 42.71 ± 4.28, $P < 0.001$), demonstrating a significant association between decreased L3-SMI and adverse outcomes in acute pancreatitis. These findings underscore the potential of L3-SMI as a valuable radiological parameter for predicting in-hospital severity and short-term prognosis in patients with acute pancreatitis (Table 2).

## Correlation between L3-SMI and MCTSI score, Balthazar CT grade

The L3-SMI, MCTSI score, and Balthazar CT grade all have significant correlations with the prognosis of severe acute pancreatitis. Specifically, L3-SMI showed a negative correlation (r = −0.640) with the prognosis, while the MCTSI score (r = 0.492) and Balthazar CT grade (r = 0.581) demonstrated positive correlations (Fig. 1). Additionally, there was a significant negative correlation between L3-SMI and both the MCTSI score (r = −0.889) and Balthazar CT grade (r = −0.790) ($P < 0.001$) (Fig. 2). This suggests that L3-SMI could potentially serve as a comprehensive representative of the MCTSI Score and Balthazar CT grade to some extent. Therefore, these findings underscore the potential of L3-SMI as an important factor in assessing the prognosis of severe acute pancreatitis.

## Evaluation value of L3-SMI for the prognosis of severe acute pancreatitis

The analysis of the ROC curve revealed that the AUC for adverse prognosis corresponding to L3-SMI was 0.816, indicating a relatively high diagnostic value. The optimal threshold value was determined to be 0.348 (with a sensitivity of 0.667 and specificity of 0.941) (Table 3 and Fig. 3).

## Multivariate logistic regression analysis of factors affecting the prognosis of acute pancreatitis

The multivariable logistic regression analysis revealed several factors influencing the prognosis of acute pancreatitis. The MCTSI score demonstrated a significant linear effect (log odds) of 0.91 ($P = 0.003$) and an odds ratio (OR) of 2.48, with a 95% confidence interval (CI) of [1.36–4.49]. In contrast, the Balthazar CT grade exhibited a linear effect of 0.53 ($P = 0.654$) and an OR of 1.11, with a 95% CI of [0.71–1.74], indicating a lack of

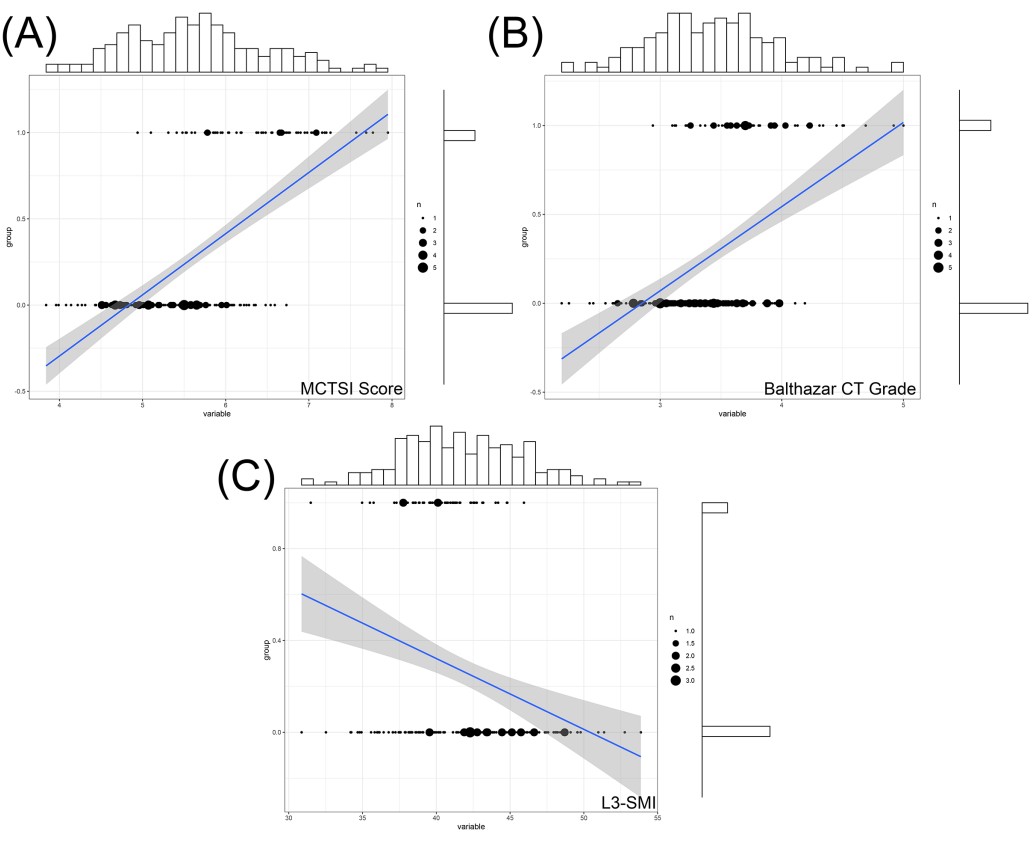

**Figure 1 Correlation between MCTSI score (A) Balthazar CT grade (B) and L3-SMI (C) within 48 h of admission with the prognosis of severe acute pancreatitis.** Histograms represents the distribution characteristics of these indexes and its corresponding prognosi.

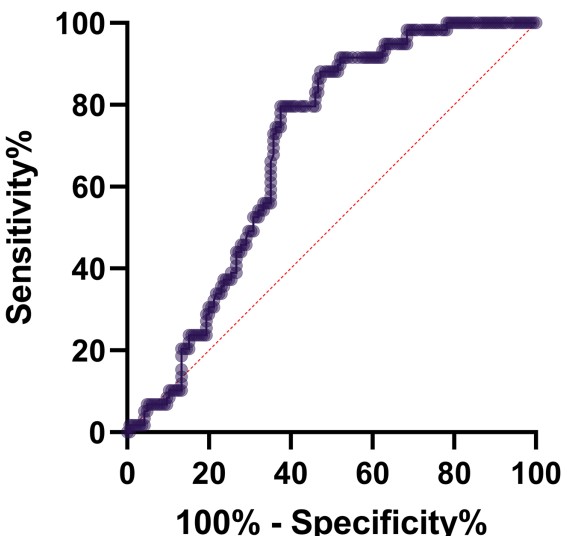

**Figure 2 ROC curve for forecasting the prognosis of severe acute pancreatitis by L3-SMI within 48 h of admission.**

**Table 3 Results of the optimal cut-off value for ROC analysis.**

| Variable | AUC | Optimal cut-off | Sensitivity | Specificity | Cut-off |
|---|---|---|---|---|---|
| L3-SMI | 0.816 | 0.384 | 0.667 | 0.941 | 0.608 |

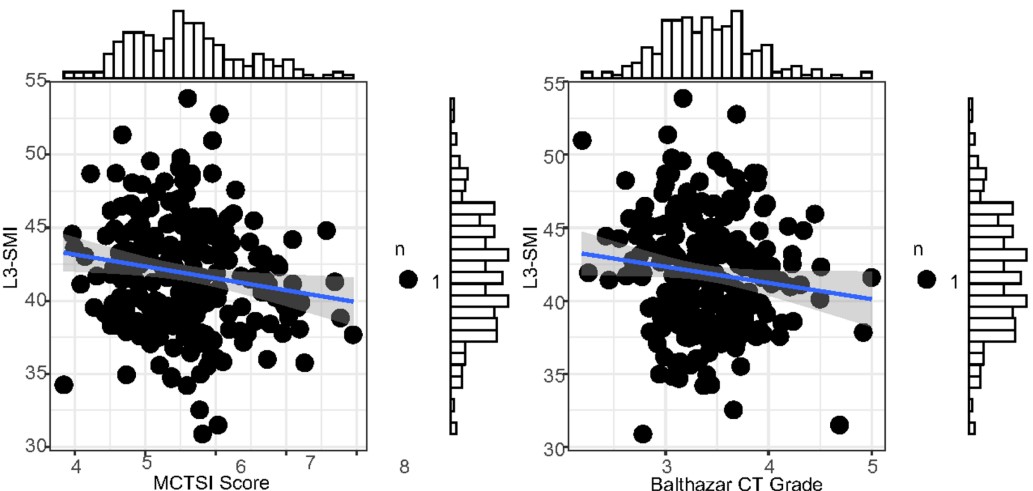

**Figure 3 Correlation analysis of MCTSI scores and Balthazar CT grading to L3-SMI within 48 h of admission. Histograms represents the distribution characteristics of these indexes.**

**Table 4 Multivariable logistic regression analysis of factors influencing the prognosis of acute pancreatitis.**

| Variables | β | S.E | Wald | Linear effect (log(odds)) | P-value | Odds ratio (OR) | 95% Confidence interval (CI) |
|---|---|---|---|---|---|---|---|
| MCTSI score | 0.91 | 0.30 | 8.89 | 0.91 | 0.003 | 2.48 | [1.36–4.49] |
| Balthazar CT grade | 0.53 | 0.23 | 5.31 | 0.53 | 0.654 | 1.11 | [0.71–1.74] |
| L3-SMI | −0.16 | 0.05 | 11.77 | −0.16 | 0.001 | 1.65 | [0.78–1.94] |
| Admission APACHEII | 0.43 | 0.31 | 1.92 | 0.43 | 0.665 | 1.14 | [0.63–2.08] |
| Mechanical ventilation | 0.69 | 0.23 | 9.00 | 0.69 | 0.399 | 1.21 | [0.78–1.88] |
| Renal replacement therapy | 0.57 | 0.27 | 4.46 | 0.57 | 0.533 | 1.18 | [0.70–2.01] |

statistical significance. The L3-Skeletal Muscle Index (L3-SMI) displayed a significant negative linear effect of −0.16 ($P = 0.001$) and an OR of 1.65, with a 95% CI of [0.78–1.94], signifying its importance as a protective factor. Conversely, the APACHE II score, mechanical ventilation, and renal replacement therapy did not exhibit statistically significant effects on the prognosis, as evidenced by their *P*-values and 95% CIs. These results underscore the critical role of MCTSI score and L3-SMI as influential factors in predicting the prognosis of acute pancreatitis, providing valuable insights for clinical risk assessment and patient management (Tables 4 and 5).

**Table 5 Multivariable logistic regression analysis of factors influencing the prognosis of acute pancreatitis.**

| Variables | β | S.E | Wald | Linear effect | P-value | Odds ratio (OR) | 95% Confidence interval (CI) |
|---|---|---|---|---|---|---|---|
| MCTSI score | 0.91 | 0.30 | 8.89 | 0.91 | 0.003 | 2.48 | [1.36–4.49] |
| Balthazar CT grade | 0.53 | 0.23 | 5.31 | 0.53 | 0.654 | 1.11 | [0.71–1.74] |
| L3-SMI | −0.16 | 0.05 | 11.77 | −0.16 | 0.001 | 1.65 | [0.78–1.94] |
| Admission APACHEII | 0.43 | 0.31 | 1.92 | 0.43 | 0.665 | 1.14 | [0.63–2.08] |
| Mechanical ventilation | 0.69 | 0.23 | 9.00 | 0.69 | 0.399 | 1.21 | [0.78–1.88] |
| Renal replacement therapy | 0.57 | 0.27 | 4.46 | 0.57 | 0.533 | 1.18 | [0.70–2.01] |

## DISCUSSION

Acute pancreatitis is a significant clinical concern due to its potential to induce multiple organ failure, consequently leading to heightened in-hospital mortality rates (*Wang et al., 2023*, *2021*; *Wei et al., 2022*). Within the purview of this study, 26.34% of the 224 patients with acute pancreatitis succumbed, corroborating prior research findings. Interventions such as surgery, mechanical ventilation, and renal replacement therapy are often necessitated in patients with acute pancreatitis (*Pinto et al., 2021*). Notably, the proportion of mechanical ventilation and renal replacement therapy among the deceased patients was relatively higher than among the survivors, a finding that is congruous with previous studies (*Padureanu et al., 2023*; *Pandanaboyana et al., 2021*). The coupling of extant literature with the outcomes of this research underscores that, notwithstanding aggressive treatments for critically ill patients with acute pancreatitis, a fraction may still relent to their severe condition. These findings accentuate the necessity for prompt recognition of early warning signs and timely interventions to mitigate the mortality rate of critically ill patients diagnosed with severe acute pancreatitis.

L3-SMI is a metric designed to quantify a patient's muscle mass by computing the ratio of muscle area at the third lumbar vertebra to body-surface area. This index functions as a barometer for patient muscle health, encapsulating aspects such as overall muscle quantity and distribution. The condition of skeletal muscle wasting, typified by a pathological decrease in systemic and progressive skeletal muscle mass, coupled with loss of muscle function, is a severe form of malnutrition frequently observed in terminal diseases such as malignancies and organ failures (*Qiu et al., 2023*). Substantiating this, *Ryan et al. (2016)* demonstrated that skeletal muscle wasting considerably amplifies adverse reactions to neoadjuvant or adjuvant chemotherapy in cancer patients, thereby attenuating treatment efficacy and significantly correlating with tumor resurgence and poor prognosis (*Ryan et al., 2016*). The use of CT has been instrumental in delivering high-quality imaging that aids in discerning muscle distribution and compositional changes. Owing to these benefits, precise computation of muscle content is feasible, rendering the calculation of muscle area at the L3 level as the most commonly utilized method for overall muscle mass prediction.

Moreover, this study unveils that L3-SMI serves as a mortality risk factor for patients with acute pancreatitis. Receiver operating characteristic curve analysis reveals that the area under the curve for poor prognosis in terms of L3-SMI is 0.816, signifying high

diagnostic value. The optimal cut-off value discerned for L3-SMI is 0.348, with an associated sensitivity of 0.667 and specificity of 0.941. The reason for this stems from the understanding that acute pancreatitis triggers inflammation of the pancreas, leading to subsequent hemorrhagic necrosis. This often escalates into a concomitant infection of pancreatic and peripancreatic necrotic tissues (*Pandanaboyana et al., 2021*). In the absence of effective infection control, this can culminate in pancreatic abscess, severe systemic infection, sepsis, metabolic aberrations, gastrointestinal dysfunction, and widespread organ damage (*Sindayigaya et al., 2022*).

Patients with acute pancreatitis experience metabolic perturbations, including elevated metabolic rate and catabolism during sepsis, which lead to a pronounced uptick in energy consumption. Hallmarks include impaired glucose utilization, escalated gluconeogenesis, glucose intolerance, insulin resistance, and elevated blood glucose levels, resulting in glycosuria (*Sykes et al., 2021*). The metabolic stress engendered by acute pancreatitis, coupled with pancreatic lipase secretion during pancreatitis, can hasten fat mobilization. This prompts a significant increase in fat breakdown or oxidative malfunctions, reflected by notable elevation in serum free fatty acids and triglycerides.

Acute pancreatitis provokes a surge in protein breakdown, particularly in skeletal muscle and other muscle tissues, culminating in significant muscle wasting. An abundance of protein breakdown products, such as urea nitrogen and creatinine, are expelled in urine, suggesting a conspicuous negative nitrogen balance (*Tennison et al., 2021*). Conversely, the synthesis of liver albumin is hampered, while the synthesis of acute phase proteins is upregulated. The serum albumin concentrations drop rapidly due to reduced synthesis, increased loss, abnormal intrabody distribution, and insufficient nutrition (*Zhou et al., 2023*).

Following these metabolic disturbances, the nutritional status of patients with acute pancreatitis deteriorates with alacrity, ushering in swift malnutrition and profound shifts in the L3-SMI index. This swift decline compromises the body's immune defense mechanisms, resulting in an escalated risk of complications, namely infections, organ dysfunction or irreversible failure, thereby boosting mortality rates (*Tucker, Hollenbeak & Goyal, 2022*; *Wyse et al., 2023*).

The APACHE II scoring system, hinging on clinical and biochemical parameters, has demonstrated proficiency in evaluating acute pancreatitis severity and prognosis. However, its complexity and intricate operationalization pose constraints (*Zheng et al., 2022*). In contrast, the Balthazar CT grading system promotes timely identification of acute pancreatitis etiology while faithfully reflecting the extent of pancreatic and peripancreatic hemorrhage and necrosis. Based on these, patients can be categorized into five tiers (0–4) (*Xu et al., 2023*). Clinical investigations have indicated that higher Balthazar CT grades correlate with more severe acute pancreatitis and elevated patient mortality rates, an observation consistent with this analysis. Thus, Balthazar CT grading exhibits a correlation with the severity of acute pancreatitis, and treatment strategies can be calibrated based on this grading and other indicators to eschew misjudgments and delayed disease progression (*Yang, Adams & Bier-Laning, 2022*).

The MCTSI supersedes the CTSI by simplifying assessment of extensive necrotic areas and peripancreatic fluid collections and incorporating evaluation of extrapancreatic complications. The elevated MCTSI scores in deceased patients compared to survivors in this study substantiate that MCTSI scoring can stratify the risk of severe acute pancreatitis to some extent, aiding clinicians in assessing disease progression and promptly adjusting treatment strategies to enhance cure rates (*Zhang et al., 2022*, *2023*).

The study discovered that deceased patients had higher MCTSI scores and Balthazar CT grades than surviving patients, showing statistically significant differences. L3-SMI levels were lower in the deceased group compared to the survival group, with a statistically significant difference. Furthermore, there was a negative correlation between L3-SMI and the MCTSI score and the Balthazar CT grade, suggesting that L3-SMI, similar to the MCTSI score and Balthazar CT grade, can be utilized in assessing the severity and short-term prognosis of acute pancreatitis in a hospital setting.

However, this study has some limitations. As a single-center retrospective study conducted at Yantaishan Hospital, the generalizability of the results may be limited. A multicenter study with a larger sample size would help validate the findings. Additionally, potential confounding factors such as smoking status, alcohol use, and medication history were not fully adjusted for in the analysis. Nutritional status is a dynamic process, but L3-SMI was only measured at admission and changes over the course of treatment were not assessed. Other markers of nutritional status and inflammation such as albumin, prealbumin, and C-reactive protein levels were not investigated. Although several prognostic variables were included in the multivariable logistic regression analysis, it is important to acknowledge that some of these variables demonstrated collinearity, as evidenced by the results of Spearman's correlation test. The presence of collinearity among these variables could potentially lead to inflated standard errors and reduced precision in estimating the relationships between each variable and the outcome. Despite the potential for collinearity, the multivariable logistic regression analysis was still able to provide valuable insights into the factors influencing the prognosis of acute pancreatitis. This highlights the robustness of the approach in accounting for collinearity and extracting meaningful prognostic information from the dataset. However, it is important to note that the interpretation of the specific effects of highly collinear variables should be approached with caution. These considerations could be addressed in future studies through advanced statistical techniques aimed at managing collinearity, such as regularization methods or dimensionality reduction, to further refine the prognostic modeling in acute pancreatitis research. While directed acyclic graphs were not explicitly mentioned in the methodology, the rationale for variable selection revolved around identifying key prognostic factors that were on the causal pathway leading to the outcomes of interest. All available variables were carefully considered based on their clinical significance and potential impact on the prognosis of acute pancreatitis. Finally, long-term outcomes beyond the in-hospital period were not evaluated. Further prospective studies with more comprehensive assessment of potential prognostic factors are needed to strengthen the conclusions regarding the role of L3-SMI in predicting outcomes in acute pancreatitis.

## CONCLUSIONS

This study corroborates the utility of diagnosing muscle depletion using L3-SMI as a valuable diagnostic tool for predicting the severity and short-term prognosis of acute pancreatitis in a hospital setting. The present findings bolster the applicability of L3-SMI in nutritional risk screening and prognosis assessment. More robust clinical investigations with expansive sample sizes and multi-center studies are anticipated to lend further validation to the clinical utility of L3-SMI. Besides, in consideration of the diverse nature of human anatomy and its impact on the assessment of the L3-SMI method, it is of great necessity for further validation and elucidation of cutoff values. Evidence of the variability in body composition among individuals is instrumental in emphasizing the need for comprehensive validation of L3-SMI assessment methods.

### Funding
The authors received no funding for this work.

### Competing Interests
The authors declare that they have no competing interests.

### Author Contributions
- Chuntao Lin conceived and designed the experiments, analyzed the data, prepared figures and/or tables, authored or reviewed drafts of the article, and approved the final draft.
- Junyuan Zhang conceived and designed the experiments, analyzed the data, authored or reviewed drafts of the article, and approved the final draft.
- Chunye Wang performed the experiments, prepared figures and/or tables, and approved the final draft.
- Wei Lian performed the experiments, authored or reviewed drafts of the article, and approved the final draft.
- Yicong Liu conceived and designed the experiments, performed the experiments, analyzed the data, prepared figures and/or tables, authored or reviewed drafts of the article, and approved the final draft.

### Human Ethics
The following information was supplied relating to ethical approvals (*i.e.*, approving body and any reference numbers):

All samples obtained in this study were approved by the ethics committee of the Yantaishan Hospital and abided by the ethical guidelines of the Declaration of Helsinki.

### Data Availability
The raw data is available in the Supplemental Files.

## Supplemental Information

Supplemental information for this article can be found online at http://dx.doi.org/10.7717/peerj.17283#supplemental-information.

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
