# Peer review of "The impact of skeletal muscle index at the third lumbar spine on nosocomial deterioration and short-term prognosis in acute pancreatitis: a retrospective observational study"

_PeerJ, doi:10.7717/peerj.17283_

## Round 0.1 · original submission · Major Revisions

Please response to the reviewers point by point.

·

Basic reporting

I think this is a grammar-wise well-written study about pancreatitis. There are a few sentences or wordings that could be changed- not deadly on line 53, rather fatal. Not death group.

I have one large methodological issue with the ROC curve. Roc curves are made for binary variables- and it is not stated how the muscle variable was made binary. This is really important.

Starting with the introduction
The first reference, I have not read it through. But it does not seem to concern primarily pancreatitis. Are there truly no better references, perhaps a couple of them, that more directly addresses pancreatitis per se. Pancreatitis is not key subject for any of the references in the reference list.
I would like to see, or have it commented that there is none, some sort of incidence or period prevalence estimate on pancreatitis in the introduction. And previous proportions of mortality.

You managed to include all the prognostic variables in the multivariable logistic regression, though some were pretty collinear, you sort of assessed collinearity when making Pearson correlations. I think you would address this in the limitations section- that the approach worked in spite of some collinearity. This is important.

Analysts are also asked not to include intervening variables and to think about this carefully before entering variables. The issue around this is frequently explored using directed acyclic graphs before analysis. (I mean this is a hot issue in epidemiologic research- how to select candidate variables for analysis.) How was this performed? Ie how was it reasoned before the study that variables were on the causal pathway, and to be analysed because of this, or did you just disregard this and analysed all available and this worked, then you could state this latter approach I think. This is important.

To me this study is rather clearly a large case series study- which is to me! also an observational study and should be conducted along the lines of how observational studies should be analysed.

Experimental design

Repeat: To me this study is rather clearly a large case series study- which is to me! also an observational study and should be conducted along the lines of how observational studies should be analysed.

Lines 85-90. In a more epidemiological study like this. The numbers in the study population are most often reported at start of results. And further, information on the number of partly eligible cases are also important, the pool from which you took your patients. For example line 117. The same physician- was he never ill or on leave? Were there perhaps some patients that were not included because this procedure was done by somebody else, was done somewhat later, or was not done. This are important information to reveal. If these patients were 100% of the group with pancreatitis this should also be stated. This is important.


56. I think these 2 abbreviations are unnecessary. Only use as few as you need.
70. You are reintroducing an already introduced abbreviation.
71 (line 71). Needs references of some sort- as does 74 to 76

79. What locale?
I think ‘death- group’ is a too blunt. A case group where patients died….or something. (line 53- fatal not deadly)
90. How was there survival defined.
.
(92. So patients did not sign informed consent?- I thought I saw this phrasing in the legal documentation. )

127. We humans are so differently built. We need some more validation information on this method and cutoffs. Ie just present what there is out there published.

147. Why don’t you define which normality test?
150 change its to related, remove it’s the second time

154. You need to be clear what variables were selected for inclusion in the multivariable methods and how you made the model building. Also mention whether you (planned to) evaluated bivariable interactions. Comes back at 184-191 where it is still unclear what variables became excluded, where tested but were reduced away. This is important.

Validity of the findings

158. Needs to start with a paragraph on the number and types of excluded patients

161. Proportion of surgical patients….1. this is not mentioned before,

173-175. Needs thorough rewriting.

180-183. I thought there should be ROC curves (perhaps in supplement) for the other scoring methods too- until I secured that we needed a binary classifier to make a ROC curve.

193-194. The first sentence I don’t understand why it is of use to state that this disease contributes to high mortality stats for an hospital…this is how it comes out to me. Sorry. Perhaps..thereby leading to high…for patients with this disease. So it only needs a slight tweaking.

The limitations section is good.

Table 1. I think the statistical methods should be found also in the tables. I have been taught that tables should be stand-alone (stressed in epidemiology).

Table 3. I would have liked to see plots also- perhaps supplement?

Table 5. Linear effects must have units included

Raw data look really good.

Reviewer 2 ·

Basic reporting

- In the abstract, please present the exact p-values instead of showing p<0.05.
- In the result section, please show the exact p-values for each statement, instead of showing p < 0.05 or p > 0.05.
- The statistics in lines 85-90 should be moved to the results section.
- Please change the “death group” to “mortality group” throughout the manuscript.
- Please spell out the full names of the tests in 1.5 statistical analysis: the name of the normality test in line 147, the name of the t-test in line 149 (e.g., two-sample independent test), the name of the z-test in line 153.

Experimental design

- In line 156, please indicate whether two-sided p-values or one-sided p-values were used to determine statistical significance.
- In Table 1, please indicate that mean and SD were used to describe the continuous variables.
- In line 153, I think the Spearman correlation test is more appropriate than Pearson’s correlation test since L3-SMI, MCTSI Score, Balthazar CT Grade are all discrete variables.
- In Table 5, looks like MCTSI score and L3-SMI are the only independent variables that are significantly associated with the dependent variables. The statement in lines 189-191 contradicts that. Please update. Please also update the relevant statements in the discussion section.

Validity of the findings

- no comment

---

## Round 0.2 · Minor Revisions

Please respond to Reviewer 1 point by point.

·

Basic reporting

You have made a number of amendments that are very good.
First I apologise for not understanding how a ROC curve can be made from logistic regression with a continuous covariate. Indeed it can. But, on the same line the answer to the question was not thorough enough. In your previous publication materials and methods lack reference or clarification how this is done. I really wanted a better explanation, e.g. what was evaluated with the z-test. But it is how it is. It is possible to google and understand this. So I am fine with this.

I don’t agree with the study not addressing that it is a case-series study. Those definitions are quite clear and can be found easily.

The first intro-sentence seems misleading to me. Googling it is easy to find less high numbers than you quote, and other diseases seem truly more common. Why do you refer to a paper about head and neck cancer? I simply don’t understand- I have not read every word of that paper, but even if it would address pancreatitis somewhere- it seems a most unsuitable source. This is still very weird.

Otherwise intro and materials and methods nice. Results- the new section at start of results is rather fleshy. But the content is fine.

Some of the new sections are too fleshy- and can be cut in half.

184. death group- cant have that. Several places

Discussion
The first sentence is taken out of the air (did we not see this previously?). I think I understand why it is there- but it is farfetched and not woven into the rest.

Figure legends lack when, where, what and how many. Apart from that I like the ROC curves. Some plots have uninformative labels, eg fig 1.

Experimental design

Think this is OK

Validity of the findings

Fine

Reviewer 2 ·

Basic reporting

The authors have adequately addressed my comments. Therefore, I have no further comments.

Experimental design

The authors have adequately addressed my comments. Therefore, I have no further comments.

Validity of the findings

The authors have adequately addressed my comments. Therefore, I have no further comments.

---

## Round 0.3 · accepted · Accept

All the comments were well addressed.

·

Basic reporting

I have no problems except one. And that is that pancreatitis is not one of the very most common clinical diseases...and I complained about this sentence. And then you fixed up the reference. But not the sentence- and I dont know how to compare. But assure me that this reference actually states that the relative importance is that huge that you are suggesting. Otherwise, and a lot more appropriate, ' P is a common clinical disease....etc etc

Experimental design

OK

Validity of the findings

OK

Reviewer 2 ·

Basic reporting

no comment

Experimental design

no comment

Validity of the findings

no comment

Reviewer 3 ·

Basic reporting

This study introduced an interesting topic on the relationship between muscle mass and prognosis among patients with acute pancreatitis. This study concluded that muscle depletion using L3-SMI is a valuable radiological parameter for predicting in-hospital severity and short-term prognosis in patients with acute pancreatitis. This finding could shed light on guidance of muscle mass in predicting prognosis among patients with acute pancretitis.

Experimental design

This study was well designed and structured.

Validity of the findings

Conclusions are well stated, linked to original research question & limited to supporting results.

Additional comments

None.